# Safe Artificial General Intelligence via Distributed Ledger Technology

Kristen W. Carlson

Department of Neurosurgery, Neurosimulation Group, Beth Israel Deaconess Medical Center/Harvard Medical School, 110 Francis St., 3rd Floor, Boston, MA 02215, USA; kwcarlso@bidmc.harvard.edu

**Abstract:** Artificial general intelligence (AGI) progression metrics indicate AGI will occur within decades. No proof exists that AGI will benefit humans and not harm or eliminate humans. A set of logically distinct conceptual components is proposed that are necessary and sufficient to (1) ensure various AGI scenarios will not harm humanity, and (2) robustly align AGI and human values and goals. By systematically addressing pathways to malevolent AI we can induce the methods/axioms required to redress them. Distributed ledger technology (DLT, "blockchain") is integral to this proposal, e.g., "smart contracts" are necessary to address the evolution of AI that will be too fast for human monitoring and intervention. The proposed axioms: (1) Access to technology by market license. (2) Transparent ethics embodied in DLT. (3) Morality encrypted via DLT. (4) Behavior control structure with values at roots. (5) Individual bar-code identification of critical components. (6) Configuration Item (from business continuity/disaster recovery planning). (7) Identity verification secured via DLT. (8) "Smart" automated contracts based on DLT. (9) Decentralized applications—AI software modules encrypted via DLT. (10) Audit trail of component usage stored via DLT. (11) Social ostracism (denial of resources) augmented by DLT petitions. (12) Game theory and mechanism design.

**Keywords:** artificial general intelligence; AGI; blockchain; distributed ledger; AI containment; AI safety; AI value alignment; ASILOMAR

## 1. Introduction

The problem of superhuman artificial intelligence ('artificial general intelligence", AGI) harming or eradicating humankind is an increasing concern as the prospect of AGI nears. This article offers a new, comprehensive set of solutions to the AGI safety problem in which distributed ledger technology (also known as "blockchain") plays multiple key roles.

We begin by citing recent significant advances in AI supporting the case that solving the AGI safety problem has become urgent. The Methods section gives the methods used to generate the axiom set proposed here and a justification for describing them at a high systems level. Other key approaches to a rigorous theory of AGI safety are suggested. The Results/Discussion section first describes the proposed axioms in some detail, referring to Appendices for detailed examples of use cases in solving exhaustive enumerations of AGI failure pathways by others, and highlights some pathways where a solution would fail without a given axiom. Two key formulae underlying the computational complexity of AGI evolution and diversity are offered, the controversial issue of restricting access to AGI technology is addressed, and metrics of AGI progress are described toward the goal of monitoring proximity to a singularity. Last, the problems of control and value alignment in successive generations of AGI, the related issue of creating a singleton versus a pluralistic separation-and-balance-of-powers approach, and using "sandbox" simulations to examine AGI safety methods are described.

Current attempts to measure AI progress show exponential growth in activity globally and technical improvement across the board of functionality measured—including "Human-Level Performance

Milestones" [1] (Figure 1a). Recent watershed advances include Deep Mind beating the most expert human at the complex game of Go—which averages 250 moves per position and 150 moves per game = $10^{359}$ possible paths vs. chess, which averages 35 moves per position and 80 moves per game = $10^{123}$ possible paths, *and a decade earlier than expected*. Deep Mind used a neural network to assign a value at each point in a decision tree and discarded low-valued lower-level branches and thus avoided the exponential search required to explore them. Human Go experts assigned high creativity to Deep Mind's strategies and tactics. A second major AI development was Deep Mind's self-teaching, reinforcement learning ability, playing tens of thousands of games against itself in a few hours rather than incorporating human game-play strategies and eliminating its need for human feedback [2].

Collaborating, self-taught AIs played 180 human years of games per day using new reinforcement learning policy optimization algorithms and beat human teamwork in the simulated real-world environment of Dota2 [3] (video: https://youtu.be/Ub9INopwJ48). Significant advances were made in credit assignment to short-term vs. long-term goals and learning the optimal balance between individual and team performance. Another watershed occurred when AI beat humans at an "imperfect information" game, poker—i.e., the opponents' hands are hidden, fundamentally different from Go or chess—using game theory techniques including bluffing, previously thought to be difficult to emulate [4,5]. Such techniques could be used to beat humans in business strategy, negotiation, strategic pricing, finance, cybersecurity, physical security, military, auctions, political campaigns, and medical treatment planning [4]. AI continues to reach new levels of unsupervised learning prowess (pattern recognition without human guidance), e.g., for parsing handwritten letters and creating new letters that pass a specialized Turing test, and more efficiently than deep learning networks [6]. AI superiority over humans in general background knowledge and parsing natural language is old news [7], and is now being embedded in all human-computer interfacing ("powered by Watson", Alexa, Siri, Cortana, Google Assistant, et al.), whose potential monetary value has triggered a commercial AI arms race in parallel with a military/political one (Figure 1) [8].

Bostrom gives examples of general intelligence skills where attainment of *any* of them would trigger AGI dominance over humans (reproduced in Table 1). One such epochal AI development that could trigger the AGI singularity is the prospect of AI learning to program itself—"recursive self-improvement" (*q.v.* ASILOMAR AI Principle #22, see also #19, #20, #21 [9])—which opens a door to a positive-feedback-driven process in which AGI vastly exceeds human capabilities in short order and may change its human-instilled directives. An AGI could begin to regard humanity as a trivial, primitive nuisance, competing for vital resources required for attainment of its goals, distinct from humanity's, stemming from alien values, as we regard mosquitoes or flies.

**Table 1.** Examples of super-intelligent skill sets triggering AGI world domination (from Bostrom [10]; cf. Babcock et al. Section 6.2 [11]).

| |
|---|
| Intelligence amplification—AI can improve its own intelligence |
| Strategy—optimizing chances of achieving goals using advanced techniques, e.g., game theory, cognitive psychology, and simulation |
| Social manipulation—psychological and social modeling e.g., for persuasion |
| Hacking—exploiting security flaws to appropriate resources |
| R&D—create more powerful technology, e.g., to achieve ubiquitous surveillance and military dominance |
| Economic productivity—generate vast wealth to acquire resources |

A danger many feared would accelerate the timeline to AGI via "Red Queen" cultural co-evolution [12], an AI arms race has begun, driven by the increasing realization in political and military circles that AI is the key to future military superiority [13,14]. Thus ASILOMAR #5 and #18 may already be violated [9]. The race increases emphasis on AI for intentionally destructive purposes and likely will result in less control of AI technology by its creators [15]. It is an ominous development

as all nuclear powers upgrade their arsenals, proliferation increases, and arms control agreements are unraveling [16]. The day when AI is consulted and decides if "no first strike" commitments or reducing "high alert" status nuclear weapons is beneficial or perceived as a vulnerable weakness by adversaries looms ahead.

The potential speed with which AGI could advance from being human-directed and empathetic of humans to evolving beyond human-level concerns is unknown; with self-programming ability or other internal intelligence enhancement, [10,11] positive feedback will trigger super-exponential growth. At that point a malevolent AGI may arise within a fraction of a second, too fast for us to detect and respond [17].

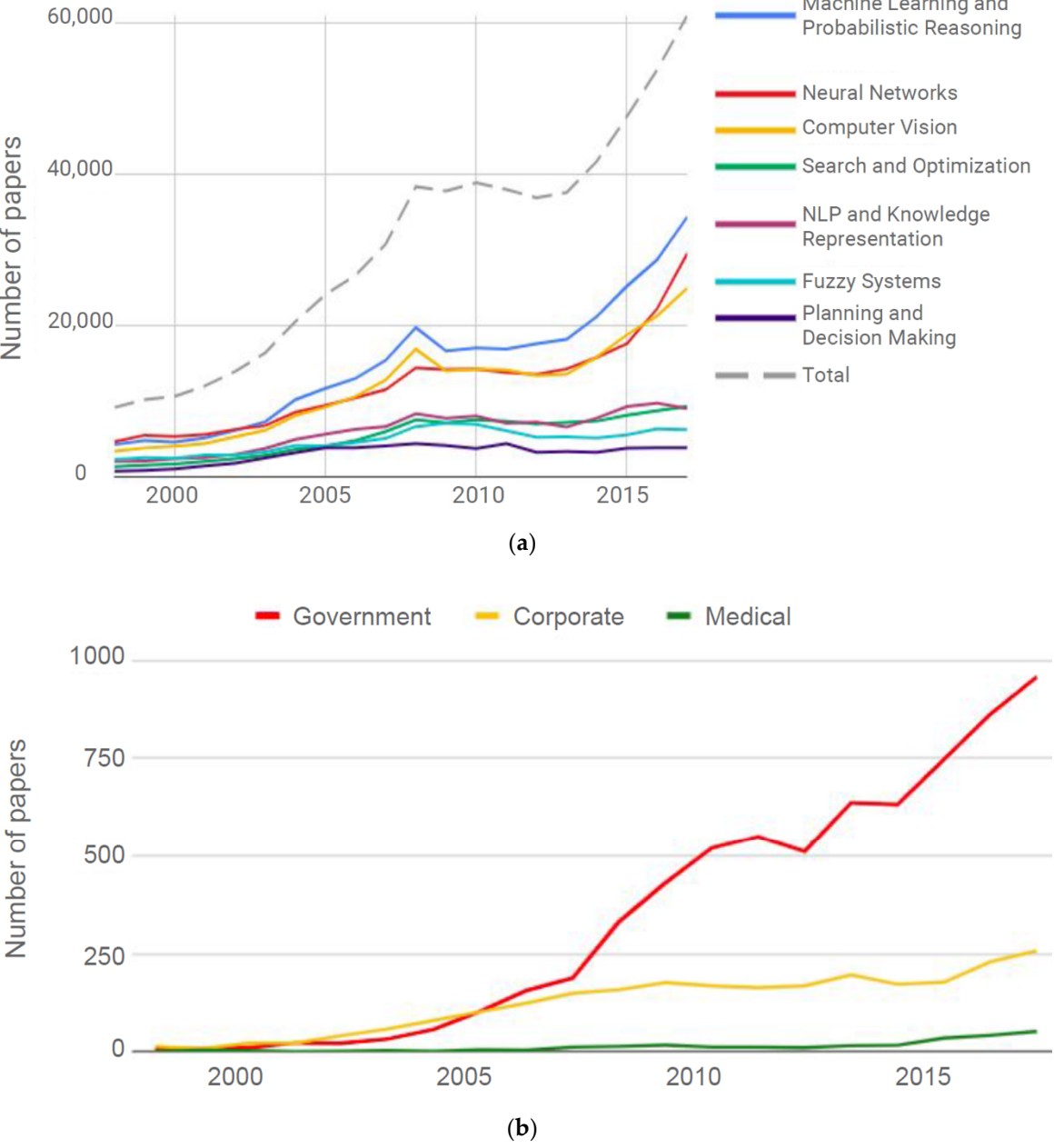

**Figure 1.** (**a**) Number of AI papers in Scopus by sub-category (1998–2017). Source: Shoham et al. {Shoham}. (**b**) Papers by sector affiliation—China (1998–2017). Source: Shoham et al. [1]. Creative Commons License. © 2018 by the authors. Submitted for possible open access publication under the terms and conditions of the Creative Commons Attribution (CC BY) license (http://creativecommons. org/licenses/by/4.0/).

What is proposed here is a complete AGI ecosystem, framed as a set of axioms at a relatively high systems level, that will ensure AGI–human value alignment, and thereby ensure benevolent AGI behavior, as seen by humans and successive generations of AGI. Notably, the axioms incorporate distributed ledger technology and smart contracts to automate and prevent corruption of many required processes.

## 2. Methods

Section 2.1 describes the methods used to generate a necessary and sufficient set of axioms for AGI safety, and comments on the feasibility of developing a rigorous proof of such an axiom set. Section 2.2 comments on the epistemology of the approach in Section 2.1, principally in terms of systems levels, and then describes other approaches that may contribute to a formal theory.

### 2.1. To Generate a Necessary and Sufficient Set of Axioms

There are several taxonomies of pathways to dangerous AI, such as Yampolskiy [18], Turchin [19], Bostrom [10], and Brundage et al. [20]. These taxonomies are a reasonable starting point for systematically investigating how to ensure safe AGI. One can take each pathway to danger as a theorem and induce methods, formalized as axioms, toward generating a necessary and sufficient set of axiom-methods to eliminate all pathways or reduce their probability. Pathway categories overlap, which helps ensure redundancy in capturing the necessary and sufficient axioms to redress all categories.

Similarly, as one iterates the process of using each dangerous pathway to generate a complete set of axioms to address it, some axioms repeat, while some pathways require new, additional axioms until at the end of the pathways list, most are covered by the axiom set, although some pathways may be left without sufficient methods to eliminate them. For the pathways itemized in the taxonomies, the resulting axioms seem to be the minimal set for ensuring safe AGI. Here "ensuring" means "optimally reducing the probability of a dangerous pathway manifesting."

Stating a set of axioms is a necessary step toward formal proof of a necessary, sufficient, and minimal set—if a formal proof is possible. Yampolskiy concludes his taxonomy by saying that formal proof of the completeness of a taxonomy is important [18] and formal methods are a main theme of Omohundro [8]. Short of a tight logical proof, probabilistically assuring benevolent AGI, e.g., through extensive simulations, may be the realistic route to take, and must accompany any set of safety measures, including those proposed here.

An important way to test if each axiom is necessary is to find failure use cases when it is omitted [21]; examples are given below.

### 2.2. Ingredients for Formalization of AGI Safety Theory

Towards formalization, the various methods to ensure safe AGI are stated as logically distinct axioms and at a high level intended to capture concisely a necessary and sufficient set. This usage of "axiom" generalizes that of von Neumann where certain lower systems level outputs or theorems are "axiomatized"—seen as black boxes, or input–output specification, or logic tables—at the immediately higher systems level [22]. Each axiom is most precisely expressed by an *operational* definition specified by an algorithm implementing it, hence, a method.

For instance, the definition of subjective value or utility, used in the morality and game theory axioms below, is made precise by the six von Neumann–Morgenstern utility axioms [23]. As stated below, a set of axioms designed, and proven via simulation, to induce cooperation among extremely diverse, complex agents may replace most of the set given herein; the simulations of Burtsev and Turchin may be prolegomena [24].

A problem we frequently face in modeling and simulation is: What is the highest systems level that can concisely describe and emulate the target set of phenomena? Thus, a limitation in axiomatic formulations is they leave varying amounts of implementation detail at the systems level underlying

them to be specified, or to some degree, developed. For example, the DLT-based axioms 2, 4, 5, 7, 8, 9, 10, and 11, are in rapid evolution toward algorithmic implementation to address diverse use cases. And behavior control (axiom 4) is in rapid development in some contexts (e.g., autonomous vehicles, factory robots), yet the degree of development still needed to align human and AGI values may be significant.

Other attempts to formalize the expression of AGI dangers are some simple syllogisms (Appendix A).

The concept of AGI-completeness, akin to NP-completeness as stated by Bostrom [10], is that a demonstration of one technology, e.g., self-improvement techniques, engendering AGI is sufficient to demonstrate that capability for a class of AI technologies. AGI-completeness may be another piece of formalizing AGI, measuring its progress, and specifying the point of no containment unless sufficient preparations have been made.

Another means to formalize AGI theory is Omohundro's idea of deriving universal AGI drives from first principles [8], which can be explored to see if such drives emerge in simulations as well as via logical derivation. Omohundro argues that universal drives will inevitably lead to conflict of AI and human values from the irrefutable economic axiom of competition for resources.

Another formalization route is calculating the probability of hacking a blockchain against the number of AGIs required to reach consensus via the blockchain to permit unlocking the next AGI generation (see sections on decentralized apps and the Singleton problem below). This calculation is similar to the math underlying the internet's redundancy in average interconnectedness of nodes and global system fault-tolerance [25] but more complicated since it involves Byzantine fault tolerance, wherein two diagnostic agents disagree on the nature of the fault [26]. The inclusion of innovative DLT into the algorithms should permit AGI robustness to surpass the "robust yet fragile" use case of the internet that is vulnerable to targeted attacks on the most interconnected nodes.

Last, it may be possible to subsume several of the axioms herein via a game theory/economics set proven via simulation. An obstacle to this approach is that game-theoretic algorithms that simulate interactions between entities with behavior expressiveness vastly larger than our own [24] may be necessary to understand and predict AGI social behavior but may also be computationally intractable (see Diversity in the AGI Ecosystem, below).

## 3. Results and Discussion

Regarding the term AI "containment", Babcock et al. suggest that "containment" is an appropriate term for methodologies for controlled AGI development and safety-testing rather than control over entities whose intelligence will exceed our own [11]. The current work is intended to contribute to both phases.

### 3.1. A Critical Ingredient: Distributed Ledger Technology (A.k.a. 'Blockchain')

The recent innovation of distributed digital ledger technology (DLT) is critical to this proposal [27]. The crux of DLT is an audit trail database, in which each addition is validated by a pluralistic consensus, currently performed by humans operating computers that run hash and anti-hash functions (to wit public key encryption), stored on a distributed network also known as a blockchain: "Blockchains allow us to have a distributed peer-to-peer network where non-trusting members can interact with each other without a trusted intermediary, in a verifiable manner" [28]. Key aspects of DLT are shown in Table 2 [29] (other auxiliary DLT aspects, such as anonymity of participants, are either not necessary or not beneficial in the context of ensuring safe AGI). The "smart" automated contract vision of Szabo [30], encrypted redundantly via DLT, could comprise the core methodology whereby AGI development and evolution can be aligned with the best human values without concomitant human intervention. Notably, smart contracts can prevent the hacking of safe AGI evolution that is too fast for human response.

**Table 2.** Distributed ledger technology applicable to ensuring AGI safety.

| |
|---|
| Non-hackability and non-censurability via decentralization (storage in multiple distributed servers), encryption in standardized blocks, and irrevocable transaction linkage (the "chain") |
| Node-fault tolerance: Redundancy via storage in a decentralized ledger of (a) rules for transactions, (b) the transaction audit trail, and (c) transaction validations |
| Transparency of the transaction rules and audit trail in the DLT |
| Automated "smart" contracts |
| Decentralized applications ("dApps"), i.e., software programs that are stored and run on a distributed network and have no central point of control or failure |
| Validation of contractual transactions by a decentralized consensus of validators |

Here are the proposed necessary and sufficient axioms to ensure safe AGI (Table 3).

**Table 3.** Proposed axioms to ensure human-benevolent AGI.

| Symbol | Axiom |
|---|---|
| 1 | Access to AGI technology via market license |
| 2 | Ethics transparently stored via DLT so they cannot be altered, forged, or deleted |
| 3 | Morality, defined as no use of force or fraud, stored via DLT |
| 4 | Behavior control structure (e.g., a behavior tree) augmented by adding human-compatible values (axioms 2 and 3) at its roots |
| 5 | Unique hardware and software ID codes |
| 6 | Configuration Item (automated configuration) |
| 7 | Secure identity via multi-factor authentication, public-key infrastructure and DLT |
| 8 | Smart contracts based on DLT |
| 9 | Decentralized applications (dApps)—AGI software code modules encrypted via DLT |
| 10 | Audit trail of component usage stored via DLT |
| 11 | Social ostracism—denial of societal resources—augmented by petitions based on DLT |
| 12 | Game theory—mechanism design of a communications and incentive system |

Table 4 gives some examples of malignant AGI categories by Bostrom [10] in which the danger pathway is described and a subset of axioms to reduce its probability is specified. To further illustrate the systematic approach of identifying a necessary and sufficient axiom set, Appendix B continues these examples using malignant AGI pathways compiled from the taxonomies of Yampolskiy [18] and Turchin [19]. In these examples, the game theory/mechanism design axiom is not mentioned; see comments in the axiom descriptions and elsewhere.

**Table 4.** Examples from Bostrom Pathways to Dangerous AI [10]. See also Appendix B.

| Pathway | Key Axioms |
|---|---|
| Perverse instantiation: "Make us smile" | Morality defined as voluntary transactions |
| Perverse instantiation: "Make us happy" | Morality defined as voluntary transactions |
| Final goal: Act to avoid bad conscience | Store value system in distributed app |
| Final goal: Maximize time-discounted integral of future reward signal | Morality defined as voluntary transactions, store value system in distributed app |
| Infrastructure profusion: Riemann hypothesis catastrophe | Morality defined as voluntary transactions |
| Infrastructure profusion: Paperclip manufacture catastrophe | Morality defined as voluntary transactions Social ostracism |
| Principal–Agent Failure [21] Human–Human: Agent (AI developer) disobeys contract Human–AGI: Agent disobeys contract | Digital identity, smart contracts, dApps, social ostracism |

*3.2. Examination of Typical Failure Use Cases by Axiom*

One way to dissect a proposed necessary and sufficient set of axioms for AI morality is to look at what phenomena or failure use cases result when one or more of them are excluded [21]; examples are given in Appendix C.

*3.3. Explanation of Each Proposed Axiom*

### 3.3.1. Access to AGI Technology via License

Two distinct systems and traditions of technology licensing exist, (1) market transactions and (2) state ("government", "fiat") coercively-controlled licensing. Seizure of AI intellectual property (IP) and control over its development by states is inevitable unless AI scientists and private-sector management set up their own systems to ensure safe AGI. ASILOMAR #9, Responsibility, states "Designers and builders of advanced AI systems are stakeholders in the moral implications of their use, misuse, and actions, with a responsibility and opportunity to shape those implications" [9]. The question is: How is this responsibility to be implemented—to be given "teeth"?

The system proposed herein envisions AI evolution with humans cross-licensing AI technology to each other, creating a prototype distributed applications (dApps) system instantiated in a DLT ecosystem that balances permissioned access and editing via contract with free access. The human-initiated DLT-based ecosystem would transition to AGIs licensing technology from humans, and subsequently to AGIs cross-licensing with each other.

History shows that in many or most cases, a market system evolves solutions faster and better than centralized state systems. Further, state systems may respond innovatively and less bureaucratically when subjected to competition with market systems; the Human Genome Project and current space-exploration efforts are examples. A market optimally distributes problems to be solved and computing power assigned to solve them in a highly decentralized manner.

There are valid arguments against an AI IP regime with "restricted" information flow via license, whether through market or state. Progress may be slowed, and some persons with no reason to be prevented from accessing some AI technology may be restricted. The counter-argument is that AGI technology and many of its components are as dangerous or more dangerous than nuclear, biological, chemical, or other mass destruction weapons technology (WMD), since AGI will control WMD tech, along with innumerable other resources that can fatally or significantly affect humanity (Proposition 1 in Appendix A).

By way of example, assume there exists an algorithm critical for AI self-programming. With free access to the self-programming algorithm, malevolent humans, as well as extant autonomous AIs, could use that technology for unlimited self-improvement, opening a positive-feedback-driven Pandora's box to unlimited malevolence and unlimited means to achieve it (ASILOMAR #22 [9]). Others point out dangers of a freely available "just add goals" AGI [10,18]. Thus state, private, or a hybrid means of restricting access to critical pieces of AI tech, as with WMD, seems to be a necessary axiom to align AI with human interests.

### 3.3.2. Ethics Stored in a Distributed Ledger

I define *ethics* as the *fundamental value system* from which autonomous entities derive their decisions and choices. *Ethics* are separate from *morality*, which is a particular set of ethics. "Honor among thieves", "do unto others as you would have them do unto you", "professional courtesy", "honor thy father and mother", etc., are ethics, as are Asimov's three laws of robotics [31]. Ethics can seem good or bad, moral or immoral, from a volitional entity's subjective value system. An entity's fundamental values are embedded in some type of behavior (input/output) control system. For example, consider ethics represented and controlled by a behavior tree [32] where the ethics are a subset of its roots, and thus in that sense *fundamental*.

The intention of storing AGI ethics via DLT is to permit a class of autonomous entities to have identical ethics and to render them visible and unable to be hacked, altered or deleted. In this sense, ethics is a necessary component of the control system and allows for different sets of ethics to be instantiated. While it is not possible for all humans to have identical values and therefore moral values (however defined), DLT, in theory, permits a universal set of immutable values to be instantiated in AGIs while still permitting an unlimited range of individual AGI and AI diversity.

Requiring transparent instantiations of ethics for AGI systems conforms to ASILOMAR #10 (Value Alignment), and IBM's call for Supplier's Declarations of Conformity for AI [33]. These *bona fides* and ethics could be stored in an AGI's Configuration Item and/or those of its key components (see below).

### 3.3.3. Morality Defined as Voluntary vs. Involuntary Exchange

The definition below is intended to conform to ASILOMAR #11, Human Values, #14, "benefit and empower as many people as possible", #15 and #24, benefit the "common good" and "widely-shared ethical ideals" [9], but notably to provide a practical implementation of them, otherwise what use are they?

Down through the ages there have been two main problems with discussions of morality—first, ambiguity and therefore confusion. How can we identify moral behavior if it is imprecisely defined and hard to determine [34]? And so such definitions are costly, in terms of the economics of law, to enforce. Second, nearly all morality descriptions are subjective, amounting to one person's value system imposed on others, and via coercion if enforced via the state.

For example, take the proposal of directing AGI to ensure "hedonistic consequentialism" for all of humanity—selecting from a set of actions the one that would produce the best balance of pleasure versus suffering [10]. Such idealistic but vague and minimally-thought-out concepts of morality—which is nearly all of them—may sound good on paper but break down rapidly on implementation. And they all amount to a minority or individual—human or AGI, and even from the most beneficent of us—deciding what is "moral" or not, or what is "best" for others. When AGI is a given, the proposals depend on its super-intelligence somehow overcoming the limitations of humans' concepts of morality, how to define and implement it, and/or overcome humans' inability to read minds. And notably, they all amount to confining computation of an overall system solution to a restricted subset of all computationally active agents (see Diversity, below), which is another way of saying allowing a subset of volitional entities to impose their subjective, not absolute, value system, upon others.

The essence of autonomy or volition is choice-making. Herein, first, all individual choices that affect no other volitional entity are moral. Second, all voluntary exchanges are moral. But if two autonomous agents prefer a transaction between them, and that transaction is prevented by a third party, that party has imposed its value system over the others. It is also one less computational experiment the entire system performs.

Several economists posited that there is no universal theory or method to determine *value*, rather, all human values and the measure of utility are subjective [35], which is implicit in the game-theoretic axioms of utility [23]. Following this premise, defining morality as all voluntary transactions is *scientific* when science is likewise defined as a procedure that filters for absolutes—what we all see in common, such as the speed of light—from a vast sea of relative views [36,37]. Later members of the Austrian school defined morality as non-interference with property (defined to include ones' body and intellectual property) [36,38]. It is simpler and less costly to define *moral* transactions as *voluntary* transactions than to try to identify what is *property* and to define and figure out property boundaries and property interference. One of the goals of a legal system is to resolve conflicts in an economically efficiently manner and it has been argued that the evolution of common law is toward such efficiency [39].

If you want to upload your mind and join a collective intelligence, or rather stay physically human, and not even accept lifespan enhancement, it is up to you. Under this system you and AGI cannot force choices on anyone else even if you or AGI believe it is best for them. But what if a super-intelligence

could make some or all of your decisions better than you can [10]? Each individual can sign on with the super-AI that seems to best fit your values and goals. It would be your choice, just like taking the advice of a consultant or hiring an agent for a specified set of tasks today.

This definition and axiom may not solve the problem of AGI with vast knowledge of the evolution of our psychology and innate choice-making algorithms [40,41] and the propensity to manipulate us with that knowledge, although the argument can be made that with such knowledge in a voluntary exchange system, AGI would be more able to offer 'good' choices (i.e., as we perceive them) to us than without that knowledge.

AGIs will have a larger and more complex set of value preferences than ours (see Diversity, below); what will be the morality of their interaction with each other? The voluntary transaction definition may fit their behavior as well. A system of voluntary transactions permits Pareto optimality and maximizes computational experiments driven by local, subjective preference systems [42]. Transaction costs and the need for trusted third parties prevents Pareto optimality [43]. DLT and smart contracts potentially permit full Pareto optimality in the digital AI ecosystem by reducing transaction costs to negligible amounts and eliminating costly, imperfect third parties.

### 3.3.4. Behavior Control System

Behavior control is sine qua non to value human–AI alignment (ASILOMAR #10, #16, etc. [9]).

At one end of the knowledge representation/control spectrum is a "flat" set of large numbers of heuristical condition–action rules that are selected, not based on general principles, but on matching specified patterns. At the other end of the spectrum is a strict postulatory–deductive tree in which the internal node "beliefs" are logically derived from the postulates as are the actions represented at the leaf-nodes. A postulatory–deductive system is the ideal contemplated here, which would satisfy the need for control, the desire for transparency of its operation, and part of the need for formal proof of its reliability. However, it is an ideal. Any type of hierarchical control system that can hold values at its highest levels and is transparent enough to reveal control over behavior by values is a candidate for aligning AGI and human values, and the ecology of value systems that will evolve from the initial sets.

I believe humans innately attempt to form postulatory–deductive systems using non-mathematical, ad hoc "logics" [40,41] in an effort to organize their world-view into causes and effects, and general principles governing specialized condition–action pairs. Mathematical and scientific postulatory–deductive systems are recent, specialized, powerful cases, improvements built on the general-purpose cognitive architecture, in which universally-valid logic replaces the ad hoc evolutionary "logics" and the entire system is validated through repeated observations directly confirming the postulates or indirectly via observation of valid derivatives (i.e., predictions) with zero fault-tolerance. Further, in the ritualized transparency of its methods and crowd-sourced validation via multiple subjective observers, science is an absolute voluntary consensus, rather than confirmation of an unprovable "objective" world [37] and resembles DLT.

In the innate human system, a causatory cascade of beliefs and actions stem from fundamental beliefs (postulates, including values). Outside of the mathematical and scientific postulatory systems, a more complex set of relative and subjective "logics" connects beliefs—efficacious from an evolutionary standpoint but also unreliable across different contexts [40,41] as seen in beliefs of mathematicians and scientists outside of mathematical and scientific domains.

An AI control system that may be able to represent current and future postulatory–deductive systems is the *behavior tree* [32].

The game-theoretic axioms of utility drive decisions from a hypothesis that the decision will ultimately lead to an improvement in the volitional entity's state, as defined internally and subjectively by its value system [23] also known as *the pursuit of happiness* [36]. The utility axioms extend to machines with subjective value systems.

### 3.3.5. Unique Component IDs, Configuration Item (CI)

Several technological and business process developments lead toward a universally interconnected system that self-configures, self-diagnoses its component failures, and repairs them automatically; in toto, a paradigm whose ultimate use will be integration into the human–AGI ecology. These technologies help to decrease Coasean transactions costs (e.g., detection and enforcement) toward facilitating an idealized Pareto-optimal economy.

Unique identification (ID) numbers evolved as an economically-efficient means to organize and validate property exchanges, contributing to a stable society, starting with large or important pieces of property such as real estate via book and page of a recorded deed, automobiles via title or vehicle ID number, stocks via CUSIP number, etc. As the cost of creating unique ID numbers decreased via technology, the system extended to machines and devices via model and serial numbers, and more recently to any product via one- and two-dimensional bar and matrix machine-readable codes to facilitate supply-chain management, quality control, customer service, and other functions.

The transition from the internet of computers to the "internet of things" (IoT) envisions ubiquitous communication and computation connecting physical devices with the digital world via miniaturized sensors and chips containing only as much computing power and energy usage that is needed to perform their intended functionality in their context—"a self-configuring network that is much more complex and dynamic than the conventional internet" [44]. In the IoT, ID numbers become digital as well as physical, e.g., radio frequency ID codes. In the IoT world AGI will be able to communicate with, and potentially control, any digital or physical device.

The IoT world was presaged by the development of *disaster recovery and business continuity planning*, and the key role of configuration items in them. Disaster recovery (DR) arose on the realization that the cost of *not* doing contingency planning for disasters (a hazardous material spill, hurricane, tornado, power outage, etc.) could vastly exceed the cost of such planning, including total business loss. Judicious planning for disasters, such as foreseeing an alternate location from which to conduct operations in the event of facility downtime and establishing redundant communication protocols to coordinate team response to disasters, are relatively inexpensive insurance measures. Business continuity planning (BCP) logically arose from DR, extending the DR premise of disaster planning to pre-planned, prioritized responses to *all* component failure, including normal end of service life. For example, recovery of failed email for the company as a whole is accorded lower priority than for customer-service representatives and top management. BCP's goal is, through contingency planning, to reduce the internal and external impact of business process downtime to a minimum.

The configuration item (CI) arose in BCP/DR conceptually as a system component's on-board algorithm and parameter set that allowed computers and components to detect each other's configuration requirements, automatically configure the component, or perform error-detection, reporting, and correction (cf. ASILOMAR #7, Failure Transparency [9] and Manheim [21]). In the context of DLT, it becomes a smart contract.

Many paths to dangerous AI, including much of the broad class of human-AI value misalignment, are a result of improperly configured or failed components, or sabotage (e.g., accidental nuclear war, failure of safeguard components, inadvertent security vulnerabilities leaving a system open to hacking, misconfiguration of software modules e.g., in autonomous vehicles, power blackouts, financial system meltdowns, etc.). Thus, the paradigm of BCP/DR and CIs will be integral to maintaining the fidelity of AGI-human value alignment amidst the IoT of the future. Further, CIs of critical AGI components can be encoded via DLT, thus greatly reducing or eliminating the possibility of unauthorized use, corruption, failure, etc.

IBM's Supplier's Declaration of Conformity to ensure AI safety [33] could be incorporated into CIs and used as one pre-requisite for deployment of an AGI system or component.

### 3.3.6. Digital Identity via Distributed Ledger Technology

Restricting access to potentially dangerous technology (Axiom #1) necessitates identity verification. Few readers would deny the need of multi-factor authentication for nuclear missile launch codes. Identity verification is currently accepted for access to military bases, high-tech weapons, aircraft, most private and public buildings, financial systems, health records, and other data that individuals consider private for their own reasons, all toward the goal of ensuring a safe and secure world.

In contrast to a third-party-based identity authentication system such as state- or private company-issued ID cards, many decentralized DLT-based methods have been created, competing with the trusted-third-party method to reduce the chance of forgery or other hacking, and bribery or other corruption. In a DLT version of the current public-key encryption-based X.509 standard [45], a DL replaces the third-party issuing authority in its components: certificate version, serial number, type of algorithm used to sign the certificate, issuing authority, validity period, name of entity being verified, and entity's public key.

Initially, digital identity verification will be done on humans matching biometrics such as facial features, fingerprint, voice, in addition to SMS etc., but as AI evolves, AGIs will use technology and techniques that they develop against evolving threats to hack verification of humans, e.g., speech synthesis or video manipulation [18] and threats that are currently unforeseeable.

### 3.3.7. Smart Contracts Based on Digital Ledger Technology

Smart contracts were conceived by Szabo decades ago, before the inventions of DLT and IoT that enable their inexpensive implementation, to automate contractual clauses via cryptography that can be self-executing and self-enforcing [46]. Smart contracts as an integral part of DLT are "scripts residing on a blockchain that automate multi-step processes" [28]. Szabo's inspirations were the original commercial security transaction protocols: SWIFT, ACH, and FedWire for electronic funds transfer, credit card point of sale terminals, and the Electronic Data Interchange for transactions between large corporations such as purchase and sale [30]. He used the simple example of a vending machine, through which transactions are performed without a third-party intermediary to verify that the terms of the transaction have been satisfied.

Two critical design goals were to make verifying satisfaction of contractual terms computationally cheap and breaching terms computationally expensive, both of which are realized in a far superior generalized manner via DLT than via prior methods (reminiscent of Bush's and Nelson's conception of hyperlinking before the invention of the internet [47]). Smart contracts require the digital specification of obligations each party must meet to trigger a transaction, a blockchain for consensus verification that each party has met its obligation, an immutable audit trail of transactions, and the design goal of excluding unintended effects on non-contractual parties.

Omohundro envisions smart contracts interfacing autonomous agents with the heterogeneity of human legal codes and future legal codes designed to help ensure safe AI interactions with humans [48] (ASILOMAR #8) [9]. Pierce envisions a mass migration of the current compliance regime via law and regulation to an economically more efficient and secure regime based on smart contracts [49] (ASILOMAR #2); such a system greatly facilitates Omohundro's.

As AGI evolves beyond our understanding and visibility, and notably when it hits "escape velocity"—exponential evolution culminating in generations succeeding each other in fractions of a second—prescribed, automated smart contracts will be essential to perpetuating ethical values in each successive generation. The concept is that a more advanced AGI generation cannot succeed a less-advanced one without licensing key components—certain algorithms, hardware, the axiom-methods proposed herein, behavior control systems invented by humans and AI, etc.—from the less-advanced generation, subject to satisfying its value system and oversight.

The configuration "handshake" between an AGI and its component CIs is a smart contract between them, and the intelligence of those handshakes can increase in the future. CIs must incorporate the ability to deny activation of a component within a system, or shut it down, if lack of satisfaction of

a given clause, or violation of a clause, of any extant contract is detected by any distributed ledger stakeholder in the transaction. All such contractual stakeholders must be silenced just as living cell cycle checkpoints must be silenced for the cell to progress through the intricately orchestrated process of mitosis, otherwise it self-destructs [50]. More of these "deadman switches" that actively suppress unauthorized use or malfunctioning AI will increase a secure evolution of benign AI; for example, the limited term of digital identity certificates that expire and require re-verification of the subject entity's identity at regular intervals [45].

Szabo's vision of embedding smart contracts in objects [30] is realized by embedding CIs in all non-trivial interconnected devices and algorithms in the IoT. In this manner the smart contract and preceding axiom-methods work in concert to ensure human-AGI value-alignment and AGI containment within bounds that are benevolent for humans and the succession of AGI generations.

In principle, smart contracts help approach a zero-transaction cost world by eliminating trusted third parties, and their role in detection and enforcement of contractual rights (e.g., physical and intellectual property rights).

### 3.3.8. Decentralized Applications (dApps)

DLT-based decentralized applications (dApps) differ from conventional application programs in that they (1) are outside the overview and control of a central authority such as a company making the app or state agency controlling it, (2) operate on a peer-to-peer network instead of a centralized one, and (3) do not have a central point of failure—they are redundant in hardware and software and therefore fault-tolerant [51]. Smart contracts are an example of dApps, as are decentralized versions of exchanges to trade various types of goods or services, notably intellectual property, which can transition into exchanges between AGIs, social media including networking, communications protocols, prediction markets, and a growing number of DLT-enabled applications.

Axiom 1, Access to Technology via Market License, requires that some dApps—notably those that are critical to AGI—would be implemented via permissioned DLs, which are DLs with an added control layer that can prevent unrestricted and unauthenticated public access. Some cryptocurrency observers feel any type of control that is not fully "public" violates the decentralization principle; however, consider "private" DLs as a critically important tool in the DLT toolbox. For example, should we not consider delegating control over access to critical AGI algorithms to a consensus of signatories committed to the goal of AI-human value alignment or ethical use of AI, e.g., the ASILOMAR AI Principles [9]? Further, the control layer, in part or eventually in toto, can be automated by incorporating smart contracts and/or smart tokens to reduce the probability that central control can be hacked or corrupted. Smart contract terms could require 2/3 or 100% acceptance of DLT-authenticated (Axiom 6) signatories to ASILOMAR AI Principles or similar regulatory documents. Smart contract terms can deny access to those who do not fulfill a transparency requirement via Supplier's Declaration of Conformity [33], which document could in turn require inclusion of an accepted set of ethics and morality (Axioms 2,3) and a safety testing record meeting certain standards [11,52], all of which can be incorporated into a CI (Axiom 5). Equally critical, dApps permit separation and balance of powers of key AGI components, analogous to no one entity having all the nuclear launch codes. The significance of dApps for ensuring benevolent AGI is discussed further in two malignant use cases it addresses, the Rogue Programmer and Singleton AGI, below.

Two levels of permissioned access to dApps may be needed: (1) Access for use, and (2) access to modify the code (while, again, a purist view of dApps sees their development as open-sourced). A similar consideration must be given to AGI technology patents. The primary purpose and requirement of patents is to "teach the art" clearly and explicitly so the innovation can be implemented by the reader. The patent system at a meta-level has largely been denied market evolution to try other purposes and requirements. Be that as it may, to facilitate *safe* free exchange of information, a "Transportation Security Administration"-type of pre-screening for access to critical AGI patents may be needed to prevent access by malevolent entities and may be efficiently implemented via smart tokens.

If no formal proof of benevolent AGI methodology is possible or available soon, sandbox simulations of new AGI technology are critical to our future and implementing them via dApps will be essential to ensure they cannot be hacked or corrupted by humans or AGIs [52].

### 3.3.9. Audit Trail of Component Usage Stored via Distributed Ledger Technology

DLT is inherently a low-cost, redundant, decentralized, hack-free audit trail—a significant improvement on traditional centralized audit trail technology. An unhackable audit trail of critical AI components such as collaborative, self-learning, or self-programming algorithms will facilitate rapid, efficient detection of their authorized or unauthorized use (i.e., a hack of a contract, a set of ethics, or an identity verification) or failure (cf. ASILOMAR #7, Failure Transparency [9] and Maheim [21]). and increase the probability of remedying the system fault. The IBM Research Supplier's Declaration of Conformity via a factsheet for AI software incorporates an audit trail as a fundamental principle [33]. Bore et al. describe a system for incorporating an audit trail in DLT as part of embedding AI simulations in DLT so that trust in the simulations' validity is enabled between researchers without requiring a trusted intermediary [52].

### 3.3.10. Social Ostracism (Voluntary Denial of Resources)

As various writers point out, a "power-hungry AGI" or "AGI pursuing world domination" implies an AGI attempting to access and control an ever-increasing amount of society's resources [10,17–19]. Therefore, the ability for entities to deny societal resources to an errant AGI is a counterforce on its ambitions. This voluntary mechanism is another aspect of a market economy in which computation is distributed, local, and optimized—each entity makes its own choice based on its own unique, subjective experience. A further optimization is that market votes can occur as often as each entity wishes to change its choice, such as denying its resources to another entity or collection of entities. Market votes occurs immeasurably more often than political votes and implement a far more fluid and asymptotically Pareto-optimal society.

In the current technology for "democracy" the political vote is the means to reach consensus, which is tallied by a central authority and enforced via coercion by the same entity. In contrast, voluntary concerted boycotts of companies, facilitated by modern social media, are increasingly affecting corporate policy (corporations being one type of voluntary association among individuals for their mutual benefit).

DLT is a fundamentally new way to reach and archive a consensus. DLT-based unhackable petitions can be smart contracts to facilitate denial of resources to an errant AGI and can be rapidly implemented via CIs. For instance, IBM's call for Supplier's Declaration of Conformity to help ensure safe AI implies voluntary adoption [33], but would be more effective if enforced via social ostracism and implemented automatically via CI incorporation, just as web browser security currently can alert a user to reject non-security-credentialed (non-https) internet domains, thereby immediately denying them the user's resources.

The ASILOMAR principles, currently signed by 1273 AI workers [9], are a significant first step, like a letter of intent, toward a necessary, more binding and important agreement. A next step could be archiving the ASILOMAR agreement and its signatories via DLT so that the principles cannot be hacked and can only be amended via consensus of the signatories. A further step could be embedding the document and signatories in the Supplier's Declaration as a second, more restricted layer of access protection. Another step would be automatically-triggered, smart contract DLT-based petitions attached to the Supplier's Declaration, denying a given set of AGI access to specified AGI technology in response to detected AGI behavior contradicting the ASILOMAR principles.

### 3.3.11. Game Theory and Mechanism Design

Game theory and evolution have explained five categories of the evolution of cooperation—direct reciprocity e.g., "tit for tat", indirect reciprocity e.g., reputation value in "what goes around, comes

around", reciprocity in societal networks and topologies, group reciprocity e.g., the good Samaritan and altruism, and kin reciprocity, e.g., "I would lay down my life for two brothers or eight cousins" (J. B. S. Haldane) [53]. Nowak's current goal is to extend these explanations to game-theoretic frameworks for global cooperation and cooperation across generations. These efforts will involve mechanism design, the branch of game theory concerned with designing game-theoretic and economic structures that build in incentives for communicating truthfully about one's valuations in a potential transaction [21,23,54,55]. That is the goal of game theory in the context of axioms for safe AGI.

It is possible that a suitably designed communication protocol and game-theoretic incentives using DLT could replace the other axioms, which would emerge from the simpler axiomatic system. For example, an axiomatic (first principles) simulation of game-theoretic evolution wherein agents have a complex set of strategies found that inclusion of two axioms, (1) inheritable agent types, and (2) visibility of types to other agents, resulted in evolution of cooperation strategies [24]. These axioms could be more general than the license, ethics, morality, configuration item, audit trail, and social ostracism axioms proposed herein. The unique component IDs, digital identity verification, and game theoretic axioms along with DLT to ensure transparency, may suffice to generate the rest of the set, just as a wide variety of market-based structures and mechanisms emerge from axiom sets that generate markets (a large proportion of economics, game theory, and agent-based modeling literature could be cited here; see, just by minimal example, the following and their references [23,54,55]).

### 3.4. Diversity in the AGI Ecosystem: Computation Is Local, Communication Is Global

However, proving this possibility may be intractable. Going back at least as far as Newell, it has been stated that the complexity of behavior (input-output functions) for $I$ inputs and $O$ outputs is $O^I$ [56]. Intuitively, this is rolling a die with $I$ faces $O$ times since any number of the $I$ inputs could map to each output. A series of actions, i.e., behaviors, is calculated by the power tower,

$$O^{I^{O^{I^{O^{I\ldots}}}}} \tag{1}$$

whose complexity grows super-exponentially. But in fact, complexity grows faster than the $O^I$ power tower in the cases where the topology of I-O mappings matters, such as in successive neural net actions. In those cases, $O$ is raised to the power set of $I$, $2^I$, and the succession of actions is calculated by the power tower,

$$O^{2^{I^{O^{2^{I\ldots}}}}} \tag{2}$$

whose complexity exceeds that of power tower 1. These intractable formulae have significant implications for the AGI ecosystem. One is that an astronomically greater diversity of value systems is possible compared to humans'. Second, AGIs' behavior in ecosystems will likely take them to disparate locations in the problem spaces they investigate, creating a very sparsely inhabited matrix of a vast number of possible behaviors. Third, in that context, game theory and mechanism design may be the key structure inducing their ongoing cooperative behavior, notably to allocate problems to be solved and communicate results that may be valuable to the other players truthfully and in a timely manner.

For example, in our primitive intellectual property regime, a protocol that induces efficient, truthful reporting is the requirement that a patent clearly teach the new art to those skilled in its subject matter. Absent that requirement and patent protection, players might be induced to seek intellectual property protection via secrecy, e.g., "trade secrets", decreasing cooperative search and overall technological progress. A protocol that induces timely reporting of innovation is the recent U. S. patent rules change to grants rights to those who are "first to file" versus "first to invent", which was economically inefficient and lacked the inducement to disclose earlier rather than later.

The fourth implication is that, as described differently in disparate intellectual settings [42,56–58], computation will continue to be performed in unique, sparsely populated loci in the general problem space using subjective criteria for exploration, and communicated via vastly shorter, high-level symbol sequences compared to the lengths of computational sequences and complexity of modeling producing them.

*3.5. Should AI Research and Technology Be Freely Available While Nuclear, Biological, and Chemical Weapons Research Are Not?*

The Rogue Programmer problem assumes that one amoral, misguided, naïve, or malevolent individual could make the single advance generating AGI, and this risk depends on how close the technology is to a single leap causing "take-off". History shows that all innovations will occur in a matter of time, some taking more time than others. For instance, differential calculus was invented by Newton in the spring of 1665 and by Leibniz in the fall of 1675 [59]. The historical record is clear that what appear in retrospect to be great innovative leaps are actually the final step built on stronger antecedents than are assumed in scientific mythology, and in fact a chain of them involving many individuals [60]. Perhaps most pertinent to the advent of AGI is the detonation of the atomic bomb by the U.S. on 16 July 1945, then by the U.S.S.R. on 29 August 1949. The fusion bomb was detonated by the U.S. on 1 November 1952 and by the U.S.S.R. on 22 November 1955, an event that was accelerated by spying, which of course is a possibility with AI research [61,62].

Such science and technology feats are large-scale group efforts. The Rogue Programmer problem arises when one individual circumvents the consensus agreement of end usage permission by the contributors to his/her technology (e.g., the 1273 AI worker signatories to the ASILOMAR principles [9]).

Two recent examples of rogue programmers are worth noting. A Chinese scientist used gene-editing techniques—developed elsewhere and made freely available in the spirit of the free exchange of ideas and technology—to change the genes of human eggs in vitro [63]. The innovation escaped overview, was motivated by ambition and pecuniary desire, and ignored a variety of the scientific community's publicly-voiced, well-thought-out *but unenforceable* concerns. Second, recently an AI programmer claimed his robot, which applied for and received citizenship in Saudi Arabia, would achieve human-level intelligence within 5–10 years [64]. His apparent variety of noble and possibly naïve motivations suggest that, even if he was not capable of making the innovation he pursues, he would combine innovations by others to achieve and claim the first human-level AI.

The problems, then, are unenforceable restrictions in a regime of "free exchange of ideas and technology", including public patents, and the lack of reliable means to measure how far away, in time or succession of innovations, we are from AGI.

*3.6. Measuring the Progression to AGI*

How urgent is the need to develop AGI-human value alignment technology? Can that debate be grounded in empirical data? Opinions differ on the timing to AGI—as of 2015 there were over 1300 published predictions [65]. Timing predictions affect the urgency of preparing AGI-human alignment and control, which influences the resources we should devote to that effort. For this and other reasons it would be helpful to measure progress to AGI in time or in successions of specific AGI-enabling technologies [66], including the positive-reinforcement, recursive self-improvement abilities such as self-teaching, collaboration, self-programming, etc.

Akin to bottom-up versus top-down economic forecasting, a method that captures and compiles many local, informed assessments is polling AI experts [65,67]. A second bottom-up approach is taken in the McKinsey Global Institute report, which assesses AI progress by its value-added to business processes using industry leader interviews and analytics [68].

A third approach, a hybrid of bottom-up and empirical metrics, is the Electronic Frontier Foundation crowd-sourcing technical progress metrics [69]. A fourth approach, empirical in concept, is taken in the AI Index 2018 Annual Report, a set of metrics intended to "ground the AI conversation in data" divided into categories: Volume of Activity, Derivative Measures, Technical Performance, Towards Human Performance, and Recent Government Initiatives and using such metrics as numbers of papers published, course enrollment, conference participation, robot software downloads, robot installations, GitHub ratings, AI startups, venture capital funding, job demand, number of patents, adoption by industry and company department, and mentions in corporate earning commentary [1].

### 3.7. AGI Development Control Analogy with Cell-Cycle Checkpoints

Biological cell division is a complex and carefully orchestrated process. Part of the insurance against cancer and other disorders resulting from defective replication is an ancient and strongly-conserved and evolved set of checkpoints that require fidelity tests to be passed in order for the cell to pass successive stages of division [50]. A notable feature of the checkpoints is their "deadman switch" setup, i.e., rather than listening for signals of defects and then emitting signals to halt the process, their default mode is to send signals that suppress entering the next stage and require active silencing by successfully passing the fidelity tests. The analogy for AGI evolution is a set of active, not passive, checkpoints that halt or delay further AGI progress until certain safety criteria established by a consensus of researchers (human or AGI) are met.

### 3.8. Intelligent Coins of the Realm

A fundamental difference between today's money and cryptocurrencies is that the latter can be "intelligent", i.e., can be endowed with more functionality than a simple token representing mutually-agreed-upon or fiat-enforced value. For example, a common AGI malevolent path is achieving world domination, inadvertently or deliberately, by commanding an exorbitant share of resources, e.g., Bostrom's paper-clip disaster [10]. Omohundro considers how universal AGI drives may be engendered and reasons that since most goals require physical and computational resources unlimited resource acquisition may be an example [8]. "Open-ended self-improvement" is another possible universal drive example [18,19]. In biological systems, cell-doubling is a potentially dangerous path to deleterious claim on resources, and cancers are a collection of such paths. It is worth noting, analogous to AGI evolution, that biological evolution has found hundreds of cancerous paths, many using re-programming to avoid cell-cycle checkpoints, and resistance to treatments is real-time exploration of new paths using various genetic algorithms [50,70,71].

As stated, the axioms provide checks, in some cases redundantly, against this danger path. An additional check and/or means of implementation could be requiring a specialized token to purchase server time or rent AGI technology that automatically looks for the requester's compliance with AGI safety agreements and standards, otherwise the requester's "credit" is denied. The token's DL then records the secure audit trail including measures of resources requested and protects against hacking to hide the evidence. Signals of possible dangerous activity, such as exponentially-increasing requests for resources by the same or related entities, could be incorporated into the token's programming. More broadly still, Omohundro cites the vision of a plethora of smart tokens performing intermediation of value and contractual obligations between the Internet of Things and humans [48].

### 3.9. The Need for Simulation of Control and Value Alignment

Considerable effort has gone into analyzing how to design, formulate, and validate computer programs that do what they were designed to do; the general problem is formally undecidable. Similarly, exploring the space of theorems (e.g., AGI safety solutions) from a set of axioms presents an exponential explosion.

A possible solution is to create a safe "sandbox" environment where, iteratively and with parameter sweeps, simulations can be performed and improvements made to *control* and *value alignment* systems until the principles resulting in robust performance validating our design intent can be induced.

Critiques of the sandbox strategy includes: (1) AGI faking benign goals or obedience in the sandbox and then pursue its actual goals when released; (2) AGI hacking out using superior technology, developed while in captivity if needed, and most generally, (3) "juvenile" AGI behavior in the sandbox that fails to predict bad behavior of a more advanced AGI into which it evolves [10]. To address #1 and #2, we need a control system that is effective enough and transparent enough to prevent those paths, such as through Axioms 2 and 3, transparent and unhackable ethics and morality, and Axiom 4, the behavior tree value system. Bore et al. take the goal of transparent simulation and modeling to a

new level by describing a system wherein simulation specifications and an audit trail are stored via DLT, thus facilitating a means to cross-validate simulations before deployment and obstruct malicious hacking or fraud in simulations by humans or AI [52] (cf. ASILOMAR #6, Safety—"verifiably so" [9]). Sandbox problem #3 may be redressed with the separation and balance of powers described next.

*3.10. A Singleton Versus a Balance of Powers and Transitive Control Regime*

Bostrom defined "singleton" as a single AGI possessing a decisive strategic advantage over humans and other AIs; a single world-dominant decision-making agency at the highest level [10]. Even if a consistent axiom set is possible that solves the AGI deception and hacking problems and others, such a set may not be sufficient to solve the problem of the singleton. The solution proposed below also addresses the proposition that ensuring *most* AGI are safe to humans is not sufficient and that *all* AGI must be rendered safe [34]. The axioms proposed herein presuppose that we cannot foresee how the evolution of AGI may outgrow the axiom set and the technology and techniques used to implement them.

Further, if simulation cannot conclusively demonstrate a solution to the singleton problem, then evolving the methods used to ensure moral, benign AGI along with AGI intelligence must be delegated to a consortium of AGIs whose values are aligned with humanity's. The idea is that a beneficent value and control system will evolve along with AGI and each generation consisting of multiple, cross-check-and-balance AGIs will, out of self-interest, endow the succeeding generation with the latest value and control version. Here "generation" means a set of AGIs incorporating a significant technological advance over a prior set of AGIs. If there is only one AGI, it seems more likely that an aberrant or errant version could emerge, while if there are, e.g., 500 AGIs in a generation that are competing pluralistically, as in markets and government based on separation and balance of powers, to win the DLT consensus to unlock the next generation-enabling AGI technology, it seems far less likely.

Thus what may lock in the transitive endowment of improved control and value alignment technology between successive AGI generations is storing the technology enabling the next generation via dApps in the blockchain and requiring multiple AGIs to reach a consensus to unlock, license, and use the tech, including control and value alignment, to succeeding AGI generations. In this manner hacking the blockchain, or attempting to coerce individual consensus agents, would be thwarted in the same way as it is done in the nascent DL methodology extant today. In addition, game theoretic design approaches may help ensure stable evolutionary strategies, likely a succession of them (dynamic equilibrium) [24,53,72]. In that context note there can be no Nash equilibrium with one overwhelmingly dominant player.

Prima facie, an entirely different way to put the principle underlying safe AGI solution to the singleton problem is to think of future AGI as a distributed automaton, and to recall von Neumann's solution to designing a reliable automaton from unreliable parts via redundancy [73]. Critical AGI algorithms may reside on multiple agents in one or more generations, who require consensus for ongoing access and cross-check each other in real time (like a deadman's switch).

## 4. Conclusions and Future Work

One epochal event likely to trigger AGI, if not the key event, is AI self-programming, or any other self-improvement, positive-feedback advancement. Close attention should be given to that development path, progress metrics and simulations developed, and measures enacted to ensure that access to key self-improvement techniques is via licensing with appropriate safeguards.

Before self-improvement technology can be unleashed, AI behavior control systems need to be developed and tested in transparent, non-hackable simulation sandbox environments as proposed by Bore et al. [52] seems essential.

If the ASILOMAR AI Principles [9] or similar agreements are akin to the U.S. Declaration of Independence, we need to move to the "Articles of Confederation", step up the current "Federalist

Papers" stage, and then move to enact the "Constitution", i.e., firm and ineluctable consensuses among leading AI workers, encrypted via DLT, as are possible.

**Funding:** This research received no external funding, but I wish it had.

**Acknowledgments:** The Seminar on Natural and Artificial Computation (SNAC), hosted by Stewart Wilson and Dave Waltz, which Stephen J. Smith and I chaired at the Rowland Institute for Science, was instrumental in my AI/machine learning education. A workshop, Can Machines Think? organized by Kurt Thearling at Thinking Machines Corporation c. 1990 catalyzed thoughts on AI safety toward a diversified, 'separation and balance of power' pluralistic system where AI agents competed to satisfy human needs.

**Conflicts of Interest:** The author declares no conflict of interest.

## Appendix A. Simple Syllogisms to Help Formalize the Problem Statement

**Proposition 1.** *Probability of Malevolent Use: With no restriction on AGI technology flow via licensing, malevolent use of AGI is a certainty.*

**Proof:** Assume: 1. There exist malevolent or incompetent humans. 2. They can freely access AGI technology (e.g., via an AGI app offering "just add goals"). Then: There will exist malevolent use of AGI.

**Corollary 1A**: With no restriction on technology flow via licensing, malevolent AGI will destroy a significant portion of humanity, or the entire species.

**Proof:** Assume in addition to 1 and 2: 3a. Some malevolent humans would employ AGI for mass destruction; 3b. Some would seek mass destruction of the entire species.

**Corollary 1B**: With no restriction on technology flow via licensing, there is a chance that malevolent AGI may destroy the entire species.

**Proof:** Assume in addition to 1, 2 and 3: 4. Some malevolent humans are incompetent in their attempts to contain their destructive goals.

**Corollary 1C:** The more widely available and easily accessible the destructive AI or AGI, the higher the probability of its deliberate or inadvertent destructive use.

**Proposition 2.** *Extent of Danger, Importance of Containing: Containing AGI is more important than containing nuclear weapon usage.*

**Proof:** Assume AGI will have control, by deliberate human consent and design, by accident, or by AGI intervention, over nuclear weapons, and in addition, other critical resources, e.g., power grid, transportation systems, financial systems, negotiations between states, etc. Then clearly AGI containment is more important than containment of nuclear weapon use.

**Proposition 3.** *Probability of Value Misalignment: Given the unlimited availability of an AGI technology as enabling as "just add goals", then AGI–human value misalignment is inevitable.*

**Proof:** From a subjective point of view, all that is required is value misalignment by the operator who adds to the AGI his/her own goals, stemming from his/her values, that conflict with any human's values; or put more strongly, the effects are malevolent as perceived by large numbers of humans. From an absolute point of view, all that is required is misalignment of the operator who adds his/her goals to the AGI system that conflict with the definition of morality presented here, voluntary, non-fraudulent transacting (Axiom 3), i.e., usage of the AGI to force his/her preferences on others.

## Appendix B. Examples of AGI Failure Modes from Turchin and Yampolskiy Taxonomies [18,19] (Continued from Table 4)

| Stage/Pathway | Necessary Axioms See Table 3 Axioms |
|---|---|
| Sabotage.<br>a. By impersonation (e.g., hacker, programmer, tester, janitor).<br>b. AI software to cloak human identity.<br>c. By someone with access. | a. 7.<br>b. 7.<br>c. 2, 3, 4, 5, 6, 8, 9, 10, 11. |
| Purposefully dangerous military robots and intelligent software. Robot soldiers, armies of military drones and cyber weapons used to penetrate networks and cause disruptions to the infrastructure.<br>a. due to command error<br>b. due to programming error<br>c. due to intentional command by adversary or nut<br>d. due to negligence by adversary or nut (e.g., AI nanobots start global catastrophe) | Axiom 3, morality, does not apply where coercive force or fraud are a premise, e.g., military or police use of force, while axiom 2, ethics, in this case embodying restrictions on use of force, and 4, behavior control, and the rest, do apply.<br>a. 1, 2, 4, 6, 8, 11<br>b. 2, 4, 5, 6, 8, 9, 10, 11<br>c. 1, 2, 4, 6, 7, 8, 10, 11<br>d. 1, 2, 4, 6, 7, 8, 9, 10, 11<br>Under some circumstances, such as if the means is already available, there is no solution (see Appendix, Proposition 1). |
| AI specifically designed for malicious and criminal purposes. Artificially intelligent viruses, spyware, Trojan horses, worms, etc. Stuxnet-style virus hacks infrastructure causing e.g., nuclear reactor meltdowns, power blackouts, food and drug poisoning, airline and drone crashes, large-scale geo-engineering systems failures. Home robots turning on owners, autonomous cars attack.<br>Narrow AI bio-hacking virus. Virus starts human extinction via DNA manipulation, virus invades brain via neural interface | 1, 2, 3, 4, 5, 6, 7, 8, 9, 10, 11<br>Under some circumstances, no solution (see Appendix, Proposition 1). |
| Robots replace humans. People lose jobs, money, and/or motivation to live; genetically-modified superior human-robot hybrids replace humans | No guaranteed solution from axiom set. All jobs can be replaced by AGI including science, mathematics, management, music, art, poetry, etc. Under axioms 1–3 humans could trade technology for resources with AGI in its pre-takeoff stage to ensure some type of guaranteed income. |
| Narrow bio-AI creates super-addictive drug. Widespread addiction and switching off of happy, productive life, e.g., social networks, fembots, wire-heading, virtual reality, designer drugs, games | 1, 2, 3, 4, 7, 8, 9, 10 |
| Nation states evolve into computer-based totalitarianism. Suppression of human values; human replacement with robots; concentration camps; killing of "useless" people; humans become slaves; system becomes fragile to variety of other catastrophes | 1, 2, 3, 4, 5, 6, 7, 8, 9, 10, 11 |
| AI fights for survival but incapable of self-improvement | 1, 2, 3, 4, 5, 6, 7, 8, 9, 10, 11 |
| Failure of nuclear deterrence AI.<br>a. impersonation of entity authorized to launch attack<br>b. virus hacks nuclear arsenal or Doomsday machine<br>c. creation of Doomsday machines by AI<br>d. self-aware military AI ("Skynet") | a. 7<br>b. 4, 6, 8, 9, 10<br>c. 1, 2 (if creation of Doomsday machine is categorized as unethical), 4, 5, 6, 7, 8, 9, 10, 11<br>d. 1, 2, 4, 5, 6, 7, 8, 9, 11 |
| Opportunity cost if strong AI is not created. Failure of global control: e.g., bioweapons created by biohackers; other major and minor risks not averted via AI control systems. | To create AGI with minimized risk and avoid opportunity cost need axioms 1–11 |
| AI becomes malignant. AI breaks restrictions and fights for world domination (control over all resources), possibly hiding its malicious intent. | 1, 2, 3, 4, 5, 6, 7, 8, 9, 10, 11<br>Note it may achieve increasing and unlimited control over resources via market transactions by convincing enough volitional entities to give it control due to potential benefits to them |

| Stage/Pathway | Necessary Axioms See Table 3 Axioms |
|---|---|
| AI deception. AI escapes from confinement; hacks its way out; copies itself into the cloud and hides that fact; destroys initial confinement facility or keeps fake version there. AI Super-persuasion. AI uses psychology to deceive humans; "you need me to avoid global catastrophe". Ability to predict human behavior vastly exceeds humans' ability. | Deception scenarios require the axioms of identity verification via DLT. Deception plus super-persuasive AI require transparent and unhackable ethics and morality stored via DLT. |
| Singleton AI reaches overwhelming power. Prevents other AI projects from continuing via hacking or diversion; gains control over influential humans via psychology or neural hacking; gains control over nuclear, bio and chemical weaponry; gains control over infrastructure; gains control over computers and internet. AI starts initial self-improvement. Human operator unwittingly unleashes AI with self-improvement; self-improvement leads to unlimited resource demands (a.k.a. world domination) or becomes malignant. AI declares itself a world power. May or may not inform humans of the level of its control over resources, may perform secret actions; starts activity proving its existence ("miracles", large-scale destruction or construction). AI continues self-improvement. AI uses earth's and then solar system's resources to continue self-improvement and control of resources, increasingly broad and successful experiments with intelligence algorithms, and attempts more risky methods of self-improvement than designers intended. | The axioms per se do not seem to solve Singleton scenarios. They are addressed in a section below where the fundamental premise is each generation of AGI will contract with the succeeding generation and use the best technology and techniques to ensure continuation of a common but evolving value system. The same principle underlies solutions to successively self-improving AI to AGI transition and AGI evolution in which humans are still meaningfully involved. |
| AI starts conquering universe at "light speed". AI builds nanobot replicators, sends them out into galaxy at light speed; creates simulations of other civilizations to estimate frequency and types of alien AI and solve the Fermi paradox; conquers the universe in our light cone and interacts with aliens and alien AI; attempts to solve end of the universe issues | The inevitable scenario where AI evolution exceeds human ability to monitor and intercede is what necessitates distributed, unhackable DLT methods and smart, i.e., automated, contracts. Further, transparent and unhackable ethics, and a durable form of morality, also unhackable via DLT, are what may ensure each generation of AGI passing the moral baton to the succeeding generation. |

## Appendix C. Typical Failure Use Cases by Axiom

| Axiom of Safe AGI Omitted from Set | Failure Use Case if Omitted |
|---|---|
| Licensing of technology via market transactions | 1. Restriction and licensing via state fiat: Corrupt use or use benefitting special interest. 2. No licensing (freely available): Unauthorized and immoral use |
| Ethics transparently stored via DLT so they cannot be altered, forged or deleted | 1. User cannot determine if AI has behavior safeguard technology (i.e., ethics) 2. Invisible ethics may not restrict moral or safe access |
| Morality, defined as no use of force or fraud, therefore resulting in voluntary transactions, stored via DLT | 1. Inadvertent or deliberate access to dangerous technology by immoral entities (human or AI), i.e., entities using AI in force or fraud 2. Note that police and military AI will have modified versions of this axiom 3. Note that this axiom does not solve the case of super-persuasive AI as alternative to fraud |
| Behavior control structure (e.g., a behavior tree) augmented by adding human-compatible values (axioms 2 and 3) at its roots | 1. Uncontrolled behavior by AGI, e.g., behavior in conflict with a set of ethics and/or morality, either deliberately or inadvertently |

| Axiom of Safe AGI Omitted from Set | Failure Use Case if Omitted |
|---|---|
| Unique hardware and software ID codes | 1. Inability for entities to restrict access to AGI components because they cannot specify them<br>2. Inability to identify causes of AGI failure to meet design intent<br>3. Inability to identify causes of AGI moral failure via identification of components causing the failure<br>Note the audit trail axiom depends on this one. |
| Configuration Item (automated configuration) | 1. Lessened ability to detect improper functionality or configuration of software or hardware within AGI.<br>2. Lessened ability to detect improper functionality or configuration of software or hardware to which AGI has access.<br>3. Inability to shut down internal AGI software and hardware modules.<br>4. Inability to shut down software and hardware modules to which AGI has access.<br>Note smart contracts and dApps axioms depend on this axiom. |
| Secure identity verification via multi-factor authentication, public-key infrastructure and DLT | 1. Inability to detect fraudulent access to secured software or hardware (e.g., nuclear launch codes, financial or health accounts).<br>2. Inability to detect AGI impersonation of human or authentic moral AGI (e.g., POTUS, military commander, police chief, CEO, journalist, banker, auditor, et al.). |
| Smart contracts based on DLT | 1. Inability to enforce evolution of moral AGI due to its pace<br>2. Inability to enforce contracts with AGI due to its speed of decisions and actions<br>3. Inability to compete with regimes using smart contracts due to inefficiency, cost, slowness of evolution, etc. |
| Distributed applications (dApps)—software code modules encrypted via DLT | 1. Inability to restrict access to key software modules essential to AGI (i.e., they could be hacked more easily by humans or AI). |
| Audit trail of component usage stored via DLT | 1. Inability to track unauthorized usage of restricted software and hardware essential to AGI.<br>2. Inability to track unethical usage of restricted software and hardware essential to AGI.<br>3. Inability to track immoral usage of restricted software and hardware essential to AGI.<br>4. Inability to identify which component(s) failed in AGI failure.<br>5. Inability to prevent hacking of audit trail.<br>6. Increased cost in time and capital to detect criminal usage of restricted software and hardware by AGI, and therefore, to apply justice and social ostracism.<br>7. Inability to compete with regimes using DLT-based audit trails due to slowness to detect failure, identify entities or components responsible for failure, and implement solutions (overall: slowness of evolution). |
| Social ostracism—denial of societal resources—augmented by petitions based on DLT | 1. Lessened ability to reduce criminal AGI access to societal resources.<br>2. Inability for entities to preferentially reduce non-criminal AGI access to societal resources. |
| Game theory/mechanism design | 1. Lacking a system to incent increasingly diverse autonomous intelligent agents to communicate results likely to be valuable to other agents and in general collaborate toward reaching individual and group goals, cohesiveness required for collaborative effort fails over time.<br>2. DLT in a digital ecosystem theoretically permits all conflicts to be resolved via voluntary transactions (the Coase theorem), but a pre-requisite set of rules may be necessary. |

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
