# Peer review of "Safe Artificial General Intelligence via Distributed Ledger Technology"

_2504-2289, doi:10.3390/bdcc3030040_

Reviewer 1 Report

The submissions described a solid research centered on the Artificial General Intelligence (AIG) paradigm impact on human beings. The author propose a set of logically distinct conceptual components that considered sufficient to 1) ensure various AGI scenarios will not harm humanity and 2) robustly align AGI and human values and goals. The reviewer considers that the targeted topics and the values of the presented outcomes makes this research work acceptable for publication. However, the following minor suggestions are considered:

Although the motivation of the presented research is clearly described at the Introduction, its main objectives must be indicated in detail. ¿how this study differs from previous publications?

The organization of the document is confusing. It should be explained at the first section of the document, and/or reconsidered.

Large tables, like Table 4 and Table 6, should be moved to Annexes

In order to enhance the scientifically soundness of the publication, singular-first persons should be avoided.

Given the nature and profundity of the insights of the conducted research, it is expected a more detailed presentation of results and future work.

Reviewer 2 Report

The author of this paper proposes complete Artificial general intelligence (AGI) ecosystem, framed as a set of axioms at a relatively high systems level, that will ensure AGI-human value alignment, and thereby ensure benevolent AGI behavior, as seen by humans and successive generations of AGI. Notably, the axioms incorporate distributed ledger technology and smart contracts to automate and prevent corruption of many required processes.

The ideas presented in the paper are very interesting and relevant to the readers of Big Data and Cognitive Computing Journal.

The manuscript is well written and focused. We only suggest the following improvements:

-      The structure of the paper should be included at the end of Introduction.

-      In Section “2. Methods” the author have the following subsection “2.1. To Generate a Necessary and Sufficient Set of Axioms” without text between them. I suggest to give a small introduction to the contents of section 2 before start subsection 2.1.

-      Line 34, correct subsection title: “3.3.5, j3.3.6. Unique component IDs, Configuration Item (CI)”

The paper can be improved if the author takes into consideration the suggestions above.
